# Galileo L10 Satellites: Orbit, Clock and Signal-in-Space Performance Analysis

**DOI:** 10.3390/s21051695

**Published:** 2021-03-01

**Authors:** Constantin-Octavian Andrei, Sonja Lahtinen, Markku Poutanen, Hannu Koivula, Jan Johansson

**Affiliations:** 1Finnish Geospatial Research Institute, National Land Survey of Finland, 02430 Masala, Finland; sonja.lahtinen@nls.fi (S.L.); markku.poutanen@nls.fi (M.P.); hannu.koivula@nls.fi (H.K.); 2Department of Space, Earth and Environment, Onsala Space Observatory, Chalmers University of Technology, 412 96 Göteborg, Sweden; jan.johansson@chalmers.se

**Keywords:** Galileo, satellite, orbit, inclination, repeat time, clock, bias, signal-in-space, availability, accuracy

## Abstract

The tenth launch (L10) of the European Global Navigation Satellite System Galileo filled in all orbital slots in the constellation. The launch carried four Galileo satellites and took place in July 2018. The satellites were declared operational in February 2019. In this study, we report on the performance of the Galileo L10 satellites in terms of orbital inclination and repeat period parameters, broadcast satellite clocks and signal in space (SiS) performance indicators. We used all available broadcast navigation data from the IGS consolidated navigation files. These satellites have not been reported in the previous studies. First, the orbital inclination (56.7±0.15°) and repeat period (50680.7±0.22 s) for all four satellites are within the nominal values. The data analysis reveals also 13.5-, 27-, 177- and 354-days periodic signals. Second, the broadcast satellite clocks show different correction magnitude due to different trends in the bias component. One clock switch and several other minor correction jumps have occurred since the satellites were declared operational. Short-term discontinuities are within ±1 ps/s, whereas clock accuracy values are constantly below 0.20 m (root-mean-square—rms). Finally, the SiS performance has been very high in terms of availability and accuracy. Monthly SiS availability has been constantly above the target value of 87% and much higher in 2020 as compared to 2019. Monthly SiS accuracy has been below 0.20 m (95th percentile) and below 0.40 m (99th percentile). The performance figures depend on the content and quality of the consolidated navigation files as well as the precise reference products. Nevertheless, these levels of accuracy are well below the 7 m threshold (95th percentile) specified in the Galileo service definition document.

## 1. Introduction

As of 15 December 2020, the European Global Navigation Satellite System, Galileo, consists of 24 operational satellites (Figure 1). The satellites were launched into space in a total of 10 launch missions: seven Soyuz launches, each carrying a pair of Galileo satellites (2011–2016) and three Ariane 5ES launches, each carrying four Galileo satellites (2016–2018). The tenth launch (L10) took place in July 2018. Satellites GSAT0219/E36, GSAT0220/E13, GSAT0221/E15 and GSAT0222/E33 were placed into slots 1, 2, 4 and 7 of the orbital plane B. As a result, all orbital slots were populated with Galileo satellites (Figure 1). After completion of all relevant commissioning activities, the four satellites were put into service in mid February 2019 [1,2,3,4].

Galileo has been offering its initial services to public authorities, businesses and citizens since mid December 2016 [5]. These services are: Open Service (OS), Public Regulated Service (PRS) and Search and Rescue Service (SAR). The three services will be complemented in the future by added-value services, namely High Accuracy Service (HAS), Open Service Navigation Message Authentication (OSNMA) and Commercial Authentication Service (CAS) [6,7]. The activation of the initial services led to an increase in the number of publications reporting on the Galileo performance in terms of ranging, positioning and timing. However, most of the studies have not concerned the most recently launched L10 satellites.

Steigenberger and Montenbruck [8] were among the first to report that the quality of broadcast orbits and clocks had improved since the beginning of the routine transmission. The researchers used data from a constellation of 12 active satellites (launches L01-L07) covering 2014–2017 period. They reported a signal-in-space range error (SISRE) of 30 cm and Galileo-only point positioning precision of 2 cm in static mode using daily solutions.

Galluzzo et al. [9] reported extensively on the Galileo status, methodology and performance metrics for the 2014–2017 time period including satellites from L01-L08 launches. The authors also concluded that the global average signal-in-space error (SISE) had improved gradually and stabilized around a constellation value of 0.50 m (95%) in 2017. In addition, the per-satellite availability was found to be above 87%, whereas 100% availability was reported for the initial open services. In terms of timing performance, the UTC dissemination error was found to be 8.9 ns (95th percentile), whereas the GPS to Galileo offset (GGTO) dissemination accuracy was estimated to be 7.2 ns (95th percentile). Furthermore, the accuracy of the on-board clock was estimated to be 0.45 m (95%) in September 2017.

Montenbruck et al. [10] reported global average root-mean-square (rms) of SISRE values at the 0.20 m level and 95th percentile values of 0.30–0.50 m for different single- and dual-frequency services. The study used data for the entire 2017 year and precise products from GeoForschungsZentrum Potsdam (GFZ) as reference.

Buist et al. [11] reported performance observed in June 2018 for ranging, timing and positioning services for a constellation of 14 satellites (L01–L08 launches). The ranging availability was found to be higher than 99.79%, whereas the worst value in the ranging accuracy was found to be less than 0.83 m for all signals (single and dual frequency) and each 14 Galileo satellites in the constellation. In addition, UTC time accuracy was found to be 8.65 ns, whereas the accuracy of the Galileo to GPS Time Offset (GGTO) was 7.91 ns for the respective month. Furthermore, the study reported accuracies below 3 m (horizontal) and 5 m (vertical) in the positioning domain.

Huang et al. [12] analysed the performance of the Galileo on-orbit satellite atomic clocks from 2014 to 2017 using the GFZ precise clock products. The phase offset, frequency and frequency drift were used to describe the clock physical characteristics. These characteristics allowed them to identify five clock switches on IOV satellites and four switches on FOC satellites before March 2017. The clock stability was evaluated to the order of $10^{-14}$ in 10,000 s.

Wu et al. [13] extended previous analyses by including all data for year 2018 (i.e., L01–L09 launches). The authors found the average rms of SISRE for Galileo constellation to improve from 0.58 m in 2015 to 0.22 m in 2018, whereas the individual annual mean values for the operational satellites varied between 0.17 to 0.29 m. Additionally, the study reported clock rms values between 0.11 and 0.21 in 2018 for the 18 Galileo satellites.

Further studies on Galileo investigated the noise level [14] or relativity effects [15] on the Galileo passive hydrogen maser satellite clocks, precise orbit and clock determination using new satellite metadata [16,17], validation on Galileo orbits using satellite laser ranging [18], impact of various precise orbit and clock products on precise point positioning [19] or the quality and the availability of real-time orbits and clocks [20,21].

Neither of the previous studies report results related to the Galileo L10 satellites. Andrei et al. [22] were among the first to report performance indicators related to the Galileo L10 satellites during their first operational year. The authors reported monthly values variations for the signal-in-space ranging accuracy from 0.17 to 0.33 m. In addition, on-board satellite clock performance derived from the precise clock products was found to be very high, with picosecond level variations and about $10^{-13}$ s/s standard deviation. Furthermore, a clock switch occurred in April 2019 and unusual variations were found in the drift rate between September and November 2019.

Recently, Alonso et al. [23] reported on the Galileo system performance, including all four Galileo L10 satellites. The study investigated an automatic detection procedure of potential faults in the satellite broadcast F/NAV navigation data, from 43 months between January 2017 and July 2020, as well as MGEX precise orbits and clocks. The authors reported results in terms of mean value, standard deviation, 68th and 95th percentiles. The overall Galileo 95th percentile statistic for the radial (0.32 m), along-track (0.55 m) and cross-track (0.34 m) orbital error components as well as satellite clock (0.34 m). The performance of the FOC satellites was found to be superior to one of the IOV satellites.

Lastly, the European GNSS Agency has published the Open Service quarterly performance reports since 2017. The reports provide the general public with information on the performance of several key performance indicators (KPIs) in the ranging, timing and positioning domains. The performance is assessed by the Galileo Reference Centre [24] with respect to the Minimum Performance Levels (MPLs) as defined in the service definition document [25]. The latest available report at the time of writing is Q3/2020 [26].

This research complements our previous results [22] on the Galileo L10 satellites, the youngest members of the Galileo constellation. This paper provides the following three contributions. First, we use all available data from the Galileo L10 satellites by the time of submission (i.e., end of December 2020). This means an 85% increase in the length of the analysed period. Second, we report on the performance of the broadcast satellite clock correction as opposed to the precise products used previously. Third, we add new investigations related to two orbital parameters: inclination and repeat period.

We found no other results on the Galileo L10 satellites except the aforementioned quarterly performance reports and the results by Alonso et al. [23]. Neither of them report on orbital inclination and repeat period. Additionally, our analysis covers the longest time span. Therefore, we believe that this study contributes independently and uniquely to the discussion related to Galileo constellation by reporting operational results to the latest Galileo satellites.

This paper is structured as follows. Section 2 describes the datasets and the analysed performance parameters. Section 3 reports the numerical results via a number of different performance metrics. Section 4 discusses the numerical findings. Section 5 summarises the key points of this paper together with the direction of the future work.

## 2. Methods and Materials

In this section, we describe the data and the methodology used to generate the performance results and analysis of the Galileo L10 satellites.

### 2.1. Datasets

Two datasets were used in this study. Both datasets were retrieved from the Crustal Dynamics Data Information System (CDDIS) portal [27]. The first dataset refers to the broadcast navigation data generated by the International GNSS Service [28] as a consolidated product. The second dataset refers to the precise orbit and clock products from the Center for Orbit Determination in Europe (CODE) [29,30] produced as part of the International GNSS Service Multi-GNSS Experiment (MGEX) project [31]. The precise products were used as a reference to assess the accuracy of the broadcast satellite clock corrections as well as to compute the Signal-in-Space ranging accuracy parameter.

The broadcast navigation data includes navigation messages transmitted by the Galileo L10 satellites since their initial activation in December 2018. Galileo satellites transmit two types of messages: Free NAVigation (F/NAV) and Integrity NAVigation (I/NAV) messages. F/NAV message is solely used by the open service, whereas I/NAV message corresponds to both open and commercial services. Table 1 summarises the navigation messages retrieved from the IGS consolidated navigation files in the Receiver INdependent EXchange (RINEX) version 3 format [32]. Both message types contain all parameters required to compute position, velocity and time (PVT) solution at the user level. One navigation message consists of ephemeris parameters, time and clock correction parameters, service parameters and almanac parameters. Figure 2 provides an illustration of the parameters that can be retrieved from these RINEX navigation files.

### 2.2. Performance Parameters

Three Galileo programme reference documents related to the provision of the Galileo Open Service (OS) are publicly available. These documents define the interface between the Galileo space segment and the Galileo user segment [33], the ionospheric model developed for the Galileo satellite navigation system and a number of key performance indicators (KPIs) along with the minimum performance levels (MPLs) of the Galileo open service [25]. The ionospheric model can be used to determine Galileo single-frequency ionospheric corrections [34]. Additionally, other figure of merits (FoMs) may be defined to independently and complementary monitor the Galileo performance at the system and/or user level. Table 2 summarises the performance parameters used in this study. They are explained in the following subsections.

#### 2.2.1. Orbital Parameters: Inclination and Repeat Period

Orbital inclination (iota, ι) is defined as the angle between the equatorial plane of the Earth and the orbital plane of the satellite (Figure 3). It is one of the six Keplerian parameters that describe the shape of the orbit, the orientation of the orbit in space and the location of the satellite along its orbit [36]. The entire procedure to convert the Keplerian parameters into Cartesian satellite positions expressed in a Earth-Centered-Earth-Fixed (ECEF) coordinate frame is given in the ICD document [33], Table 58.

The computation of the inclination angle requires several parameters from the broadcast data. The following mathematical formula is applied:(1)$$\iota =\iota_{0}+\dot{\iota }\times \left(t-toe\right)+\Delta \iota$$
where: ι denotes the inclination of the orbital plane at time *t*, $\iota_{0}$ denotes the inclination angle at reference time (toe), $\dot{\iota }$ denotes the rate of change of the inclination angle, toe denotes the reference time of the ephemeris data, Δι denotes the inclination correction due to the harmonic sine/cosine coefficients (Cis,Cic), argument of perigee (ω), and the true anomaly (ν).

Orbital repeat period (*T*) is defined as the time a satellite takes to complete one revolution around the Earth. The computation of the orbital period requires two parameters from the broadcast navigation data: the square root of the semimajor axis and the correction to the mean motion.
(2)$$T=2\pi \frac{1}{n}=\frac{2\pi }{\sqrt{\frac{GM}{a^{3}}}+\Delta n}$$
where: *n* denotes the mean motion, GM denotes the Geocentric gravitational constant, *a* denotes the semimajor axis, Δn denotes the correction to the mean motion.

#### 2.2.2. Satellite Clock Correction

Satellite clock correction is needed to align the satellite clocks to the system time in order to determine the precise moment in time when the navigation signal was transmitted by the satellite. Following the ICD document [33], we estimated the correction using the following mathematical equation:(3)$$\delta t^{SV}=a_{f0}+a_{f1}\times \left(t-toc\right)+a_{f2}\times {\left(t-toc\right)}^{2}+\delta t_{rel}$$
where: $a_{f0},a_{f1},a_{f2}$ represent the clock correction parameters (bias, drift, drift rate) transmitted in the navigation message at a specific epoch in time, *t* denotes the Galileo time for which the user computes the clock correction, toc denotes the reference time for the clock correction, $\delta t_{rel}$ denotes the relativistic correction term.

Both the difference t−toc and the relativistic correction are expressed in seconds. The latter is computed using the following equation:(4)$$\delta t_{rel}=F\times e\times \sqrt{a}\times sin\left(E\right)$$
where: $F=-2\frac{\sqrt{\mu }}{c^{2}}$ is a function of the geocentric gravitational constant and the speed of light. Both of them are constant numerical values (Table 59 [33]). As a result, the following value is used:


$$F=-4.442807309\times 10^{-10}s/m^{1/2}$$



 *e* denotes the orbit eccentricity, $\sqrt{a}$ denotes the square root of the semimajor axis, *E* denotes the eccentric anomaly.


The satellite clock correction always refers to a dual frequency signal combination. Therefore, one can distinguish two type of clock corrections: E1/E5a clock correction as derived from the F/NAV navigation data and E1/E5b clock correction as derived from the I/NAV navigation data. Our study reports on the E1/E5a clock correction to be in line with the definition of the precise product used to assess the accuracy.

In addition, Section 3.2 reports the results on the rate of change or discontinuities over time for the E1/E5a clock corrections using the following simple first order derivative to handle the uneven distribution of data over time.
(5)$${\dot{\delta t}}^{SV}=\frac{\delta t^{SV}\left(t_{2}\right)-\delta t^{SV}\left(t_{1}\right)}{t_{2}-t_{1}}$$
where: $t_{1},t_{2}$ denote two consecutive epochs for which the satellite clock correction parameter is estimated.

Furthermore, the study reports the differences between the satellite clock corrections with respect to the precise reference product using the root mean square (rms) accuracy metric. This metric gives a relatively high weight to large errors, if present.

#### 2.2.3. Signal in Space Availability and Accuracy

The service definition document [25] defines a number of KPIs in the ranging, timing and positioning domains. The KPIs in the ranging domain refer to the signal characteristics, such as status, availability and accuracy. Our study reports on two official KPIs: Signal-in-Space (SiS) availability and accuracy.

SiS availability per individual satellite is defined as the amount of time that a specific satellite transmits a healthy signal. The availability is expressed in percentage of time and includes planned and unplanned outages. The signal status refresh rate is typically between ten minutes and three hours. After four hours without updates, the transmitted navigation data is considered expired, and it should not be used in the Galileo services [25].

SiS ranging accuracy is defined as the 95th percentile of the time series of the global average Galileo SISE. The accuracy is measured only for the time periods during which the satellite transmitted a healthy signal [25]. SISE provides the instantaneous difference between the broadcast and reference Galileo satellite position and clock corrections, projected on the user-satellite direction. The broadcast values are obtained from the navigation message, whereas the reference values are taken, for example, from the IGS CODE final product [30]. Section C.4.3.2 in the service definition document [25] gives the following approximation equation:

(6)$$\mathrm{SISE}=\sqrt{0.96910\times R^{2}+{\mathrm{CLK}}^{2}+0.01545\times \left(A^{2}+C^{2}\right)+1.96881\times \mathrm{CLK}\times R}$$
where *R*, *C* and *A* denote the differences in the satellite position along the radial, cross and along directions, whereas CLK denotes the difference in the satellite clock corrections. The coefficients provided in Equation (Equation 6) come under the assumption that an elevation masking angle of 5 degrees is applied by the Galileo receivers. SISE was computed using in-house software developed at Chalmers.

Additionally, Section 3.3 reports also the 99th percentile statistic metric. This additional metric helps to gain more comprehensive understanding on the performance by complementing the officially defined metrics. Furthermore, the additional metric helps to identify possible issues in the consolidated navigation file or other possible slips in the computation procedure. Both the 95th and 99th percentile statistics are derived from SISE values at 5-min intervals to match the precise products used as reference values. They take into account only the navigation data that are marked “Healthy”.

## 3. Results

In this section, we report the results related to orbital inclination and repeat period and satellite clocks. We also analyse the performance in terms of SiS availability and accuracy. The reported results depend on the content and quality of the broadcast navigation file as well as the precise products.

### 3.1. Orbital Parameters

Figure 4 illustrates the time series of the orbital inclination parameter for all four Galileo L10 satellites from December 2018 to December 2020. The time series reveal a 0.249°/year linear trend. The inclination has increased about 0.48° in the last two years. However, the values are within the nominal value of 56 ± 2° [35]. The detrended time series shows periodical changes. They are in the order of ±0.03°.

Figure 5 illustrates the numerical results for the orbital period parameter derived as explained in Section 2.2. The mean value over the study period is 50,680.7 s, with a standard deviation of 0.22 s. The peak-to-peak variations are in the range of 1.50 s (GSAT0219/E36, GSAT0220/E13) and 1.52 s (GSAT0221/E15, GSAT0222/E33).

Our results show that it takes on average 14 h 04 m 40.7 s for a satellite to complete a revolution around the Earth. This operational revolution period is on average 1.2–1.3 s shorter than the nominal period of 14 h 04 m 42 s [35]. It suggests that the satellites might operate on orbits with slightly shorter semimajor axis in comparison to the theoretical value of 29,600.318 km [35]. The one second change in orbital period translates approximately to 200–250 m change in the semimajor axis.

The orbital period parameter time series are clearly periodical by nature. They have both lower and higher frequencies included. The lower frequencies have the typical form of annual plus semiannual signal. The higher frequencies can be better seen in the zoomed in plots in Figure 6. The top plot displays the entire time series for year 2020, the middle plot displays a three months period from May to July 2020, whereas the bottom plot displays one month period, i.e., May 2020.

In order to study the periodical signals, we downsampled the unevenly distributed 10 min data into daily data. The time series are too short to robustly estimate the yearly periods, thus we analysed also a few other satellites from the same orbital plane with roughly four years of data to support the findings. The analysis showed peaks for 354 and 177 days. These correspond to the lunar yearly and half-yearly periods. We fitted a deterministic model to the daily average time series using the Hector software package [37,38]. The model consists of linear trend and two periodical signals as depicted in Figure 7. The estimated amplitudes, summarised in Table 3, are on average 0.124 and 0.137 s for the 354- and 177-day periods, respectively. The results show also 0.1 s/y trends. These are probably related to several year periods. However, the time series is very limited to drawing solid conclusions on the long trend and variations.

The analysis also shows peaks for 13.5 and 27 days. In addition, there are further daily and sub daily variations visible in Figure 6. These variations are related to the satellite position with respect to the Earth, Moon and Sun system. It is rather challenging to interpret these variations due the complexity of the system. Nevertheless, these findings are further discussed in Section 4.1.

### 3.2. Satellite Clock Corrections

These results are different from [22] where the analysis was carried out on the satellite clock corrections derived from the IGS CODE precise product. Nevertheless, the same precise products are used here to assess the accuracy metric for the broadcast satellite clock corrections.

Figure 8 illustrates the satellite clock corrections derived from the broadcast navigation data after the satellites were declared operational (see Table 1). The plot on the left shows the corrections computed as explained in Section 2.2. The corrections are different in magnitude and in the order of hundreds of microseconds. GSAT0221/E15 shows the largest positive values, whereas GSAT0220/E13 shows the smallest. The corrections also display different behaviour over time. A downward trend can be seen for GSAT0219/E36 and GSAT0221/E15, with the trend being more pronounced for the first satellite. An upward trend can be seen for GSAT0220/E13. The plot also shows a big jump in the clock correction for GSAT0222/E33. This is associated with the maintenance activities in the beginning of April 2019, when the Galileo operator switched the signal transmission to a different clock [39]. Here, it is important to point out that each Galileo satellite is equipped with four atomic clocks: two Rubidium Atomic Frequency Standards (RAFSs) and two Passive Hydrogen Masers (PHMs). This level of redundancy was chosen to comply with the lifetime requirements of the Galileo satellites [40] Any of these clocks may be used in the signal generation, although PHM type is the most common due its superior performance [14,15].

The plot on the right shows the corrections after removal of the linear trend (i.e., linear least-squares fit to data was subtracted). No breakpoints are considered. This plot shows smaller magnitude correction jumps and changes in trends over different periods of time. In addition, the detrended values show higher order effects related to the changes in the satellite clock drift parameter.

Figure 9 illustrates the distribution of the first-order differences of the satellite clock corrections as derived from the broadcast navigation data. We used Equation (Equation 6) to compute the 1 s discontinuities since the time series of the computed corrections is unevenly spaced. The distribution shows that 99% of these discontinuities are within ±1.6 ps/s, whereas 99.99% are within ±5.5 ps/s. They give an indication of the stability of the broadcast clock corrections over time. Furthermore, we also computed the accuracy of the broadcast clock corrections with respect to the CODE precise product [30] to gain further understanding on the quality of the broadcast satellite clock corrections.

Figure 10 depicts the root-mean-square (rms) error between the broadcast and precise clock corrections. The annual rms value indicates a magnitude around 0.20 cm for both 2019 and 2020. Additionally, the monthly rms values are also consistently below 0.20 cm. However, there are few cases when the rms values are larger than 0.20 cm. These cases are further discussed in Section 4.2. It is anticipated that none of these cases led to a significant degradation of the ranging accuracy, which was well below the 7 m threshold (95th percentile) specified in the Galileo Service Definition Document [25].

### 3.3. Signal in Space Performance

Figure 11 depicts the SiS availability for all signals transmitted by the four Galileo L10 satellites from March 2019 to December 2020. Only the complete months after being declared operational are included. The results for 2019 were explained in detail in our previous research [22]. Nevertheless, we recall here that the main reasons for the lower availability numbers were the planned outage (April 2019), the Galileo incident (July 2019) and transmission of No Accuracy Prediction Available (NAPA) events during a specific month (e.g., November 2019). The latter were the most frequent and accounted for about 2.5% of the transmitted messages. NAPA is an indicator of a potential anomalous SiS [33]. A NAPA event changes the status of the signal from healthy to marginal.

There is a clear enhancement in SiS availability from all four satellites in 2020 compared to 2019. The GSAT0221/E15 stands out among the quadruplets with 100% SiS availability for all satellite signals in 2020 except in December. This was in connection with the service degradation event on all Galileo satellites on December 14. During the degradation period, the status of all satellite signals changed to “Marginal” for about 4 h (on F/NAV) and 6 h (on I/NAV) starting from the midnight. The nominal service on all satellites was resumed starting with 07:00 UTC same day [41]. Perfect SiS availability was also recorded from the other satellites except four months: March 2020 (GSAT0220/E13), June 2020 (GSAT0222/E33), September 2020 and November 2020 (GSAT0219/E36). The reason for the reduced availability was the transmission of NAPA values. GSAT0220/E13 transmitted NAPA values on I/NAV navigation message for about one hour on March 14, whereas GSAT0222/E33 transmitted NAPA values on both F/NAV and I/NAV navigation messages on June 11. Moreover, GSAT0219/E36 transmitted NAPA values on both F/NAV and I/NAV navigation messages for almost 24 h between 1–3 September. Furthermore, the same satellite transmitted NAPA events on F/NAV navigation message for about 7.5 h on two different days, November 1 and 15, as well as on I/NAV navigation message for about 4.5 h on November 15. Nevertheless, the monthly availability was higher than 93.50% for all satellites and signals in 2020.

Figure 12 depicts the accuracy metric in terms of two statistical indicators that characterise the time series of the Galileo SISE for the E1E5a signal.

The plot on the left shows the 95th percentile of the time series of the global average SISE. This statistical indicator represents one of the Galileo key performance indicators, i.e., Galileo ranging accuracy [25]. The plot on the right shows the 99th percentile of the same time series. This indicator is not official but a FoM defined for this study. Both indicators are computed on daily and monthly basis. However, the figure displays only the monthly values.

The 95th percentile plot shows a monthly indicator in the range of 0.20–0.30 m. The monthly values vary between 0.169 to 0.329 m. GSAT0219/E36 performed constantly the best among the quadruplets, whereas the GSAT0220/E13 performance was on the higher side of the range, with a difference in the order of a decimeter. Individually, the best performing months were March 2019 with 0.174 m (GSAT0219/E36), June 2019 with 0.241 m (GSAT0220/E13), September 2019 with 0.169 m (GSAT0221/E15) and July 2020 with 0.185 (GSAT0222/E33). Overall, June 2020 was the best month with all four satellites performing better than 0.242 m. Furthermore, three of the satellites recorded the best daily performance in 2020, with values below 0.10 m. The best performing days were in 3 September with 0.083 m (GSAT0219/E36), 2 April with 0.085 m (GSAT0222/E33) and 0.094 m (GSAT0221/E15). The fourth satellite (GSAT0220/E13) recorded its best daily value of 0.124 m on 15 December 2019. Overall, the best day for the quadruplets was 2 September 2020 when all four satellites performed better than 0.166 m. Table 4 summarises all these statistical numbers. This statistical indicator was well inside the 7 m minimum performance level specified in the Galileo OS Service Definition Document [25].

The 99th percentile plot also confirms the consistent performance. In addition, it demonstrates the robustness of the performance levels. This indicator was constantly in the range of 0.20–0.40 m. Overall, the best month was June 2020 when all satellites performed better than 0.310 m, whereas the best day was recorded on 5 September when all quadruplets performed better than 0.179 m. The best monthly value for a single satellite was 0.240 m in July 2020 (GSAT0221/E15). Otherwise, monthly values higher than 0.40 m were recorded in eight different cases: four correspond to GSAT0222/E33, two correspond to GSAT0219/E36, whereas there is one case each for the remaining two satellites. There are five cases in 2019 and three cases in 2020.

The cases in June 2019 and November 2020 are explained by large errors (3–18 m) in the orbital and clock components for two consecutive days: June 3–4 and November 15–16, respectively. The cases in August 2019, November 2019, May 2020 and October 2020 are due to medium errors (1–5 m) in the orbital and clock errors for multiple days: August 1–3, November 26–27, May 22–23 and October 22–23, respectively. The cases in July 2019 (GSAT0221/E15, GSAT0222/E33) are prior to the Galileo incident [42,43]. Medium errors (around 1 m) only in the clock component occurred during July 11. For some of the cases, the signal status changed to “Marginal” after or during the degradation period. The epochs marked “Marginal” were discarded from the statistic computation. However, for the other cases, the signal status remained “Healthy” during the degradation period. SISE was computed only for the time periods during which the transmitted signal was marked “Healthy”. This is in line with the “conditions and constraints” indicated in the service definition document [25] for the computation of the minimum performance level associated with the ranging accuracy. As a result, the degradation periods are visible in the 99th percentile. Nevertheless, all values in the 99th percentile plot are still well below 7 m level defined as minimum performance level corresponding to a lower percentile (95th).

## 4. Discussion

### 4.1. Time Series Analysis of the Orbit Inclination and Orbital Period

The orbital inclination has a positive trend that seems to be associated to the 35.7-year precession of the orbital plane [44]. The study points out that although the inclination returns to its original value after one cycle of 35.7 years, the inclination angle will be subject to variations of ±2°. The detrended time series show short and medium periodic oscillations. These oscillations might reflect the frequencies of both the third-body (Sun and Moon) and the satellite [45]. In addition, the orbital period is strongly correlated with the semimajor axis, a Keplerian parameter transmitted in the navigation data. Thus, most of the variations in the length of the orbital period may express the noncentrality of the Earth’s gravitational potential and the gravitational pull of the Moon and the Sun [45]. Furthermore, inclination can be changed by the force perpendicular to the satellite orbital plane due to the shape of the Earth. Galileo satellites have 10:17 resonance orbits. Thus, higher order terms of the Earth potential might have some influence, but their magnitude is much smaller than e.g., the GPS satellites that have 1:2 resonance orbits.

### 4.2. How the Broadcast Satellite Clocks Performed?

The satellite clock corrections suggest good broadcast clock performance all four Galileo L10 satellites. The findings presented here are in line with Andrei et al. [22] results that used the precise clock correction product.

The broadcast clock corrections for the individual satellites differ in magnitude due to different trends in the bias component. One clock switch and several other minor correction jumps have occurred since the first day of operation. Short-term variations in the correction value are found to be within ±1.6 ps/s (Figure 9). This level of variation is slightly larger then the ±1 ps/s rate-of-change derived from the precise clock product [22]. Furthermore, the accuracy of the broadcast clock corrections expressed as rms error is found to be around 0.20 m, with annual value in 2020 slightly better than 2019. This level of accuracy is consistently obtained for the individual monthly values (Figure 10). The results are similar to other Galileo satellites launched before and reported in previous studies.

In terms of robustness, the monthly rms values reveal two interesting aspects. The first aspect relates to all satellites in October 2019. Initially, the statistics differed significantly from the other months. This is explained by the presence of a large constellation mean due to another Galileo satellite. On 29 October 2019, the IOV satellite GSAT0101/E11 experienced several hundred meters large satellite clock errors between 18:00 to 18:45. Afterwards, the satellite was marked “Marginal” and thus removed from the nominal service. Without any anomaly or fault detection algorithm, the large error propagated in the approach applied in this study via the constellation mean to all other satellites. After removal of the faulty data, the results are in line with the other months. Please refer to Montenbruck et al. [10] for further information on how and why a constellation mean is removed.

The second aspect is related to individual satellites. The rms error value for GSAT0222/E33 in June 2019 is almost twice larger than for the other satellites. This is explained by large (up to 5 m) broadcast clock correction error on 3 June. The error magnitude decreased to 1–2 m on the next day until 09:45 a.m., when the status changed to “Marginal”. The satellite experienced the same behaviour in May 2020 although the clock errors were in the range of ±1.5 m during 22–23 May. Additionally, clock error in the range of ±3 m was found for GSAT0219/E36 satellite in 15–16 November 2020 despite the status changed to “Marginal” only between 10:00 and 15:00 on 15 November. Although SISE computation does not require it, these findings indicate the need of an anomaly and/or integrity detection algorithm to be able to remove any potential faulty satellite when computing performance metrics. This becomes even a more significant issue when estimating the position, velocity and/or time at the user level.

Our results related to the Galileo L10 satellites are also in an excellent agreement with a recent study published in the end of November 2020 by Alonso et al. [23]. The study uses gAGE consolidated navigation files and MGEX precise products. It includes navigation until 31 July 2020. Thus, Galileo L10 satellites are also included albeit with shorter coverage period than our study. However, the Alonso et al. study does not report rms values but the overall mean, standard deviation, 68th and 95th percentiles. To compare the two studies, we also cross checked these statistics. We found subcentimetre agreement between the two studies.

Two more aspects to point out here. First, the quality of the broadcast clock corrections is highly dependent on the update interval of the navigation data. Under normal circumstances, the update interval varies between 10 and 80 min, and occasionally it goes up to 180 min. The update rates are much higher (i.e., updated more often) than the two hour in the case of GPS. The high update rates and the quality of the broadcast clocks are attributed to the superior stability of the on-board clocks, in particular PHMs [14,15,46]. Second, the reported performance of the broadcast clocks also depends on the data quality in the consolidated navigation files as well as precise products. We found no documentation publicly available on how the navigation files are generated. Moreover, there are differences among precise products from different IGS Analysis Centres [17,19,47]. Steigenberger and Montenbruck [47] reports orbit consistency around one decimetre for the Galileo orbits and about 5 cm for the Galileo clocks. On the other hand, Prange et al. [17] report that orbit accuracy within the Galileo constellation is very homogeneous in the CODE MGEX solution although Galileo orbits are slightly degraded during eclipses. Therefore, our findings might also be influenced by the heterogeneous quality of the input data.

### 4.3. How Was the Signal-in-Space Performance in Terms of Availability and Accuracy?

The numerical results reveal that Galileo L10 satellites have complied with the target values in terms of SiS performance. SiS availability was considerably higher than 87% for most of the months with some exceptions all occurring in the first operational year. One hundred percent availability was achieved for at least 14 out of the first 21 operational months. A clear increase in the SiS availability is noticeable in the second operational year (i.e., year 2020) as compared to the first one (i.e., year 2019). The availability figures are in line with the ones reported in the quarterly performance reports [26].

The high signal availability is complemented by a robust and consistent performance from all four satellites in terms of accuracy. Both statistical indicators, 95th and 99th percentiles, were consistent and at comparable levels despite some minor differences among the satellites. The 95th percentile statistic, which defines the Galileo OS ranging accuracy KPI, demonstrates that the performance is well within the 7 m minimum performance level defined for Galileo satellites [25]. Galileo L10 satellites have consistently and robustly reached 0.20 m SiS accuracy, performance level comparable with previously launched satellites [48].

In 2020, the best performing months were recorded during the summer months in June, followed closely by July and August, whereas the best performing day was recorded on 5 September. It comes shortly after 30 August when the maximum daily value was 0.201 m for all four satellites. Although the officially quarterly reports do not state satellite specific values, our findings are in line with the ranging accuracy at constellation level (over all satellites) reported for Q3/2020.

These unprecedented levels of performance demonstrate the reliability, robustness and consistency of the signal transmission for the youngest members of the constellation. Presently, Galileo outperforms other GNSSs. According to Montenbruck et al. [48] the SiS performance is explained by the use of highly stable on-board clocks (Passive Hydrogen Massers) that facilitate clock offset prediction as well as uplink capabilities that refresh the navigation messages at intervals of 10–100 min.

## 5. Conclusions

We presented results on the operational performance of the Galileo L10 satellites in terms of orbital inclination and repeat period parameters, broadcast satellite clocks and SiS performance indicators. We used all available broadcast navigation data from the IGS consolidated navigation files. Our numerical analysis showed that the common reported parameters were consistent and in good agreement with the quarterly performance reports issued by the Galileo Service Center. We conclude our main findings as follows.

First, the Galileo L10 satellites have been operating within the nominal orbital parameters. The orbital inclination shows a clear positive trend with a mean rate of 0.249°/year. The orbital repeat period indicates short, medium and long oscillation patterns at various intervals (13.5-, 27-, 177- and 354-day). These periodic oscillations reflect the frequencies associated with the satellite, Earth, Sun and Moon system.

Second, the broadcast satellite clock corrections for all four Galileo L10 satellites have been in good agreement with the CODE precise clock product. This is attributed to the superior stability of the on-board Galileo Passive Hydrogen Masers (PHMs) and the high update rates of the navigation data. We found these rates to be typically between 10 to 80 min, although occasionally the update rate switched to the 180 min automatic rates. The satellite clock annual and monthly rms values have been consistently below 0.20 m. However, for some months this threshold was surpassed due to presence of medium or large outliers in the broadcast satellite clocks for some days in that specific month. None of these outliers led to a significant degradation of the ranging accuracy. Therefore, we recommend the need of an anomaly and integrity detection algorithm to identify potential faults in the satellite data before being used in any statistic computation or in estimating the position, velocity and time parameters at the user level.

Third, the Signal-in-Space operational performance (availability and accuracy) from the four Galileo L10 satellites has complied with the target values defined in the Galileo Service Definition Document. The SiS availability improved consistently in 2020 compared to 2019. One of the quadruplets (GSAT0221/E15) stands out with 100% SiS availability for all signals in 2020. The monthly SiS ranging accuracy is in the 0.20–0.30 m range (95th percentile). There are numerous days when this performance level goes even below 0.10 m. Overall, the best month was June 2020 when all four satellites performed better than 0.24 m, whereas the best daily performance was 2 September 2020 when all four satellites performed better than 0.17 m. The robustness and consistency of the performance is enhanced by an analysis on the 99th percentile indicator. This analysis also allows for the identification of several special cases affected by outliers despite the navigation data being marked valid and status is “Healthy”. The outliers are caused by medium and/or large errors in the SISE orbital and clock components. Although SISE definition does not specify any outlier removal procedure, we recommend an outlier screening procedure at the user level to detect those periods of time when the broadcast navigation data may not match the desired performance levels.

Finally, we would like to point out that our results are based on third party IGS products. These products were obtained in general about two weeks after the end of the previous month. We noticed that some of the broadcast consolidated files had been updated over time. No data quality and/or other integrity checks have been carried out on the Galileo navigation parameters retrieved from the consolidated navigation files. The same applies for the reference precise orbits and clocks. Therefore, our results do not necessary reflect issues related to the Galileo system and its performance.

In terms of future work, the research will naturally be extended to all satellites in the Galileo constellation. Other performance indicators, such as User Equipment Error (UEE) and User Equivalent Range Error (UERE) are also to be considered in the future.

## Figures and Tables

**Figure 1 sensors-21-01695-f001:**
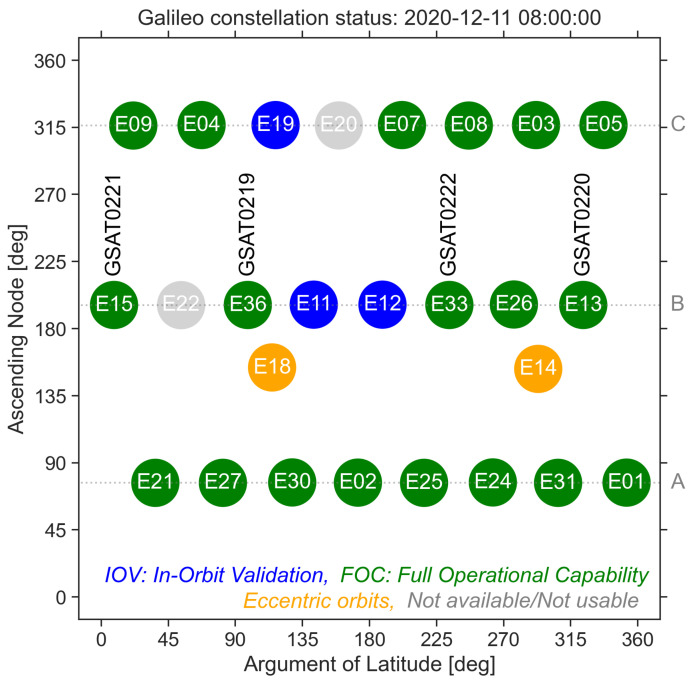
Position of the Galileo L10 satellites within the entire Galileo constellation. The satellites are arranged in three orbital planes depicted by the capital letters (A,B,C) on the the right hand side. Different colours are used to denote the type of the satellites: IOV, In-Orbit Validation (blue), FOC, Full Operation Capability (green), satellites in eccentric orbits (orange) and not available/not usable satellites (grey).

**Figure 2 sensors-21-01695-f002:**
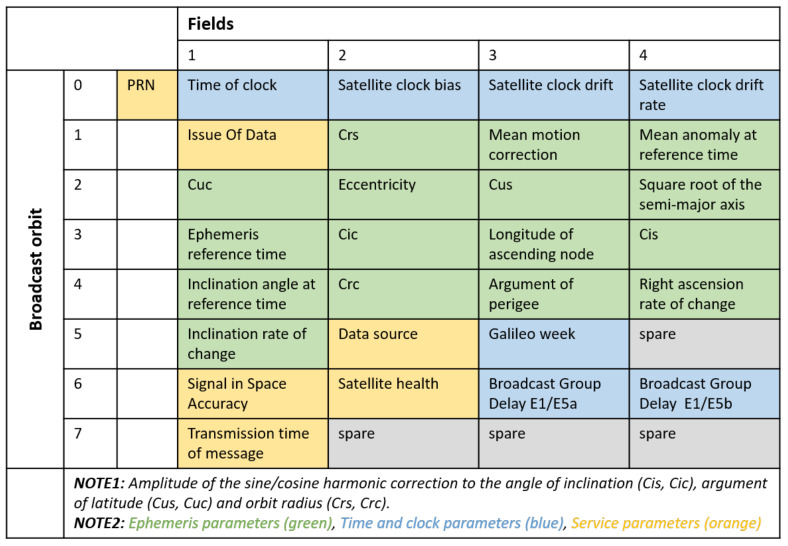
Parameters retrieved from a Galileo navigation data record as found in the BRDC00IGS product provided in the RINEX version 3 format [32]. Please notice that some parameters are navigation message specific. Thus, the numerical values are different between F/NAV and I/NAV navigation message.

**Figure 3 sensors-21-01695-f003:**
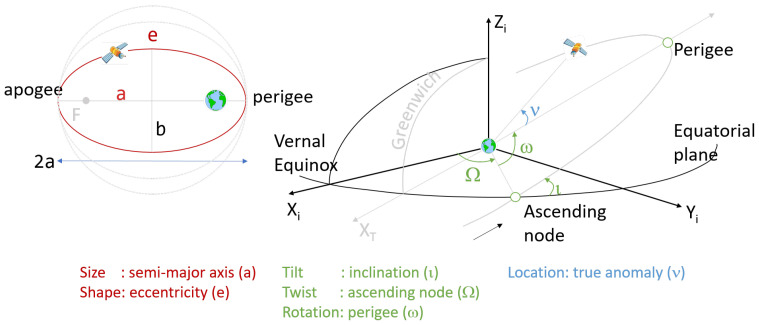
Keplerian parameters to describe the satellite position in space.

**Figure 4 sensors-21-01695-f004:**
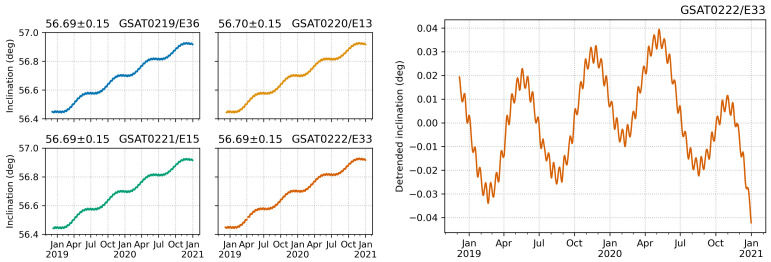
Orbit inclination evolution of the Galileo L10 satellites from the beginning of the navigation data transmission to December 2020: original values (**left**) and detrended values (**right**). [Unit: degrees].

**Figure 5 sensors-21-01695-f005:**
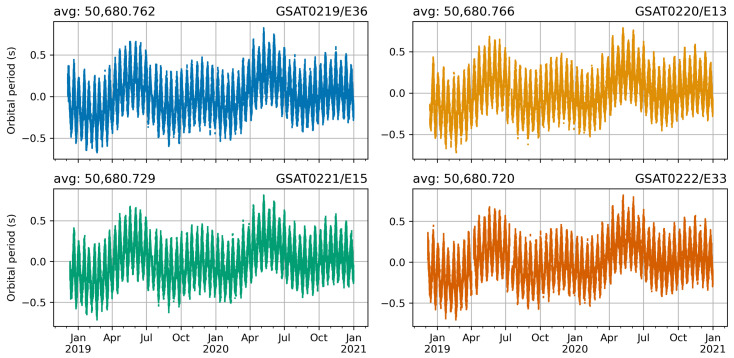
Orbital repeat period variations of the Galileo L10 satellites from the beginning of the navigation data transmission to December 2020. [Unit: seconds].

**Figure 6 sensors-21-01695-f006:**
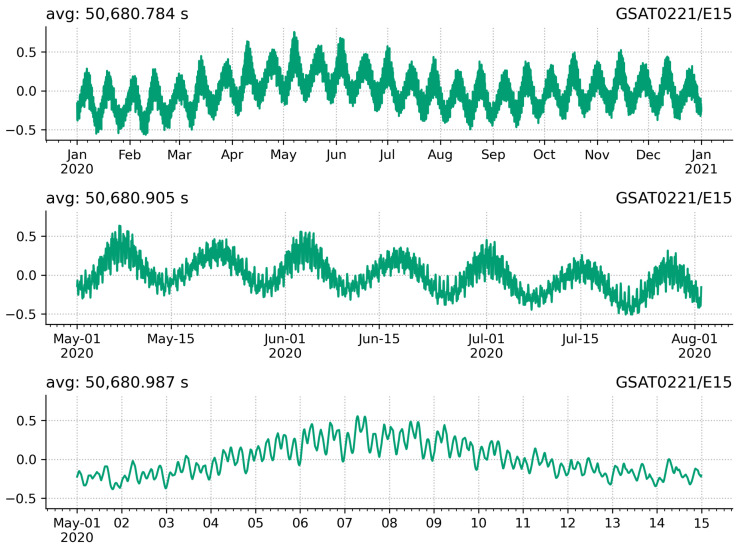
Periodic variations in the orbit repeat period over different periods of time. The **top** plot displays the variations for year 2020. Time marks for the 7th, 14th, 21st and 28th day of each month in 2020 are also given for guidance. The **middle** plot displays the variations over a three month period from May to the end of July 2020. Daily tick marks are also plotted. The **bottom** plot displays the variations during May 2020. Marks for every four hours within a day are also plotted. [Unit: seconds].

**Figure 7 sensors-21-01695-f007:**
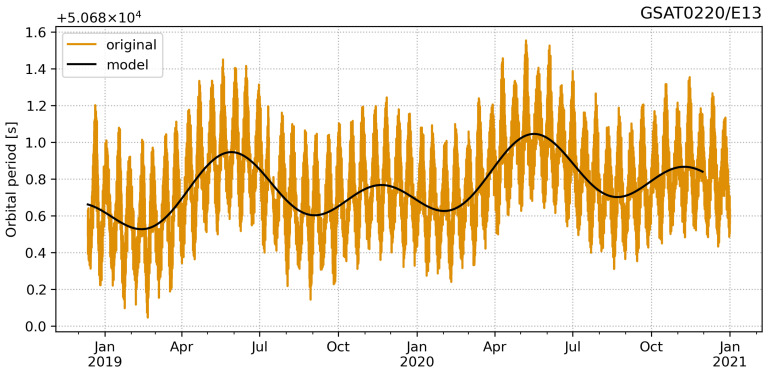
GSAT0220/E13 orbital period time series and the fitted trajectory model consisted of linear trend and two periodic signals (177- and 354-day period). [Unit: seconds].

**Figure 8 sensors-21-01695-f008:**
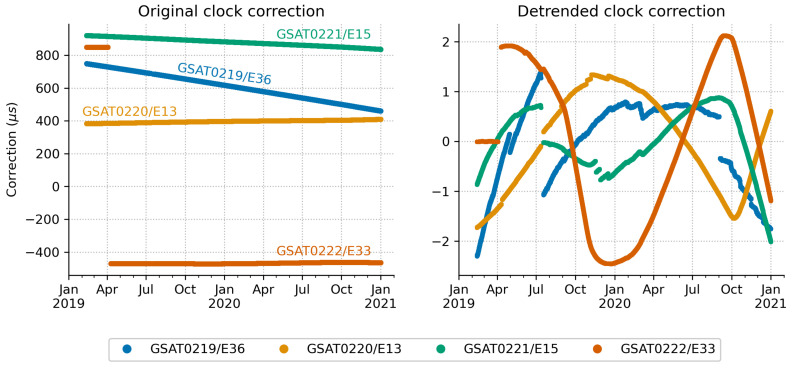
Satellite clock corrections as derived from the broadcast navigation data after the satellites were declared operational. The **left** plot displays the magnitude of the corrections. The **right** plot displays the same data after removing a linear-squares fit. No breakpoints were applied. It reveals several jumps in the correction magnitude as well as changes in trend over time. [Unit: microsecond].

**Figure 9 sensors-21-01695-f009:**
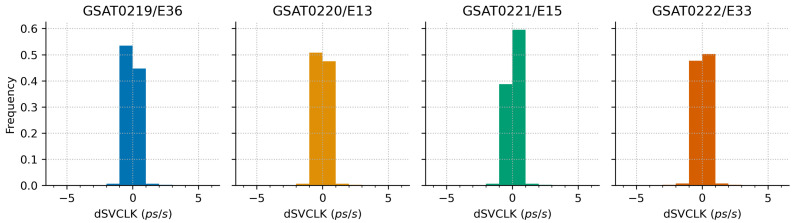
First-order discontinuities in the satellite clock corrections as derived from the broadcast navigation data using Equation (Equation 5). [Unit: picosecond/second].

**Figure 10 sensors-21-01695-f010:**
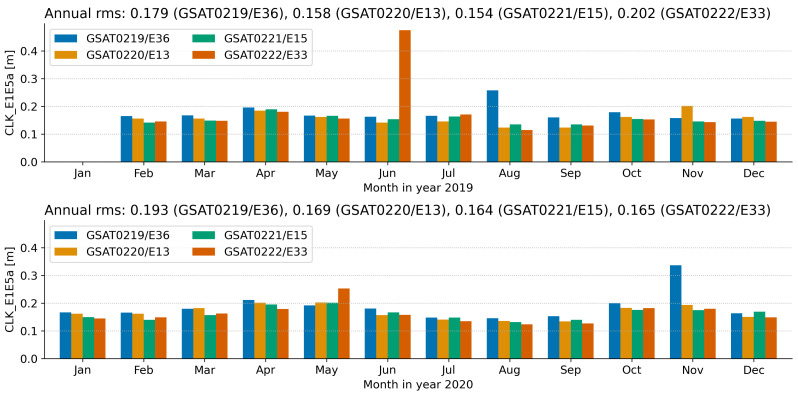
Accuracy of the broadcast satellite clock corrections as annual and monthly root mean square error (RMSE). The **top** plot displays the accuracy for year 2019, whereas the **bottom** plot displays the accuracy for year 2020. RMSE is computed with respect to CODE precise clock product. In addition, a constellation bias at each epoch is removed from the difference. Refer to Section 4.2 for further discussion on the results. [Unit: meter].

**Figure 11 sensors-21-01695-f011:**
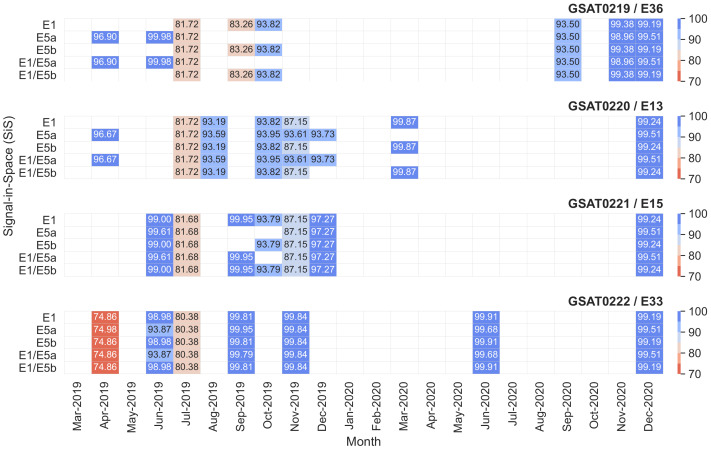
SiS availability per-satellite transmitting a healthy signal from March 2019 to December 2020. Availability is computed starting with the first complete month after the satellites were declared operational. The values lower than 100% are shown numerically for guidance. A white cell denotes 100% SiS availability. [Unit: percentage].

**Figure 12 sensors-21-01695-f012:**
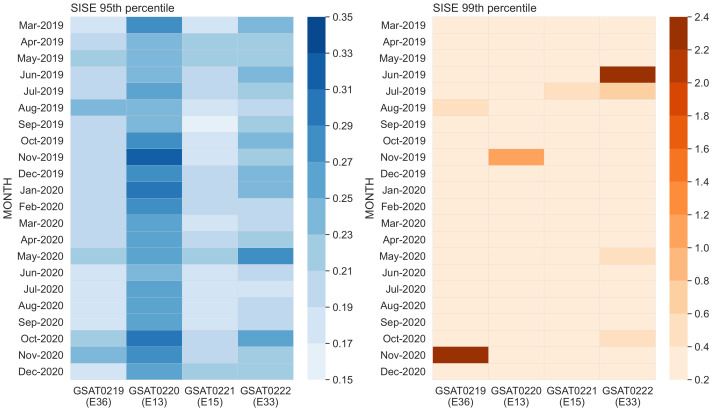
Signal-in-Space ranging accuracy expressed through two statistical indicators: 95th percentile (**left**) and 99th percentile (**right**) of the global average SISE time series for E1E5a signal. [Unit: meter].

**Table 1 sensors-21-01695-t001:** Navigation messages transmitted by the Galileo L10 satellites over the study period. Each satellite has its unique space vehicle identification number (SVID). In addition, each satellite transmits a specific pseudo random number (PRN) code.

SVID	PRN	Navigation Message		Time of Ephemerides		Operational
	Code	FNAV	INAV	First	Last	11–Feb–2019
GSAT0219	E36	64691	64521	04-Dec-2018 18:10:00	31-Dec-2020 22:40:00	10:26:00 UTC
GSAT0220	E13	58298	58346	12-Dec-2018 02:10:00	31-Dec-2020 23:40:00	10:56:00 UTC
GSAT0221	E15	64390	64349	10-Dec-2018 17:30:00	31-Dec-2020 23:40:00	11:26:00 UTC
GSAT0222	E33	60174	60102	06-Dec-2018 20:30:00	31-Dec-2020 23:40:00	12:10:00 UTC
TOTAL		247553	247318			

**Table 2 sensors-21-01695-t002:** Performance parameters and their associated metrics used in this study.

Nr.	Performance	Type	Performance	Target	Reference
crt.	Parameter		Metric	Value	
1	Orbit inclination	FoM	mean, std.	56 ± 2°	Table 9.1, [35]
2	Orbit repeat period	FoM	mean, std.	14 h 04 m 42 s	Table 9.1, [35]
3	Satellite clock correction	FoM			
	(a) rate of change		1st order derivative	-	this study
	(b) accuracy		root-mean-square	-	this study
4	Signal in Space availability	KPI	% of time	≥87%	Table 13, [25]
5	Signal in Space accuracy	KPI	95th percentile	≤7 m	Table 9, [25]
		FoM	99th percentile	-	this study

**Table 3 sensors-21-01695-t003:** The estimated amplitudes and trends for the orbital period derived from the daily means time series.

Parameter	Amplitudes (s)		Trends (s/y)
Satellite	354-Day Period	177-Day Period	
GSAT0219/E36	0.121 ± 0.038	0.139 ± 0.037	0.09 ± 0.05
GSAT0220/E13	0.121 ± 0.038	0.133 ± 0.037	0.10 ± 0.05
GSAT0221/E15	0.125 ± 0.038	0.136 ± 0.037	0.10 ± 0.05
GSAT0222/E33	0.128 ± 0.039	0.138 ± 0.038	0.10 ± 0.05
Mean	0.124 ± 0.038	0.137 ± 0.037	

**Table 4 sensors-21-01695-t004:** Signal in Space Error (SISE) global average best month and best day [Unit: meter]

Statistics	GSAT / PRN	Best	Value	Best	Value
		Month		Day	
95th	GSAT0219/E36	Mar-2019	0.174	03-Sep-2020	0.083
percentile	GSAT0220/E13	Jun-2019	0.241	15-Dec-2019	0.124
	GSAT0221/E15	Sep-2019	0.169	08-Aug-2020	0.094
	GSAT0222/E33	Jul-2020	0.185	02-Apr-2019	0.085
	quadruplets	Jun-2020	0.242	02-Sep-2020	0.166
99th	GSAT0219/E36	Jun-2020	0.250	03-Sep-2020	0.088
percentile	GSAT0220/E13	Apr-2019	0.304	10-Apr-2019	0.129
	GSAT0221/E15	Jul-2020	0.240	07-Sep-2020	0.116
	GSAT0222/E33	Jul-2020	0.255	05-Jun-2019	0.130
	quadruplets	Jun-2020	0.310	05-Sep-2020	0.179

## Data Availability

All data used in this study are produced as part of the International GNSS Service Multi-GNSS Experiment (IGS MGEX) project [31]. The data are openly available from the Crustal Dynamics Data Information System (CDDIS) portal [27]. The broadcast consolidated navigation files are generated by the International GNSS Service [28]. The precise orbit and clock products are produced by the Center for Orbit Determination in Europe (CODE) [29,30].

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
