# Peer review of "Galileo L10 Satellites: Orbit, Clock and Signal-in-Space Performance Analysis"

_sensors, 2021, doi:10.3390/s21051695_

Round 1
Reviewer 1 Report
In the paper titled “Galileo L10 satellites: orbit, clock and signal-in-space performance analysis” the authors report on the performance of Galileo L10 satellites in terms of two orbital parameters, broadcast satellite clocks, and signal in space (SiS) performance indicators based on all broadcast navigation data available.
The article title is accurate and concise. In the entire paper, the authors use standard technical and scientific terminology. The manuscript is well written and structured.
The idea of the research is interesting and presents some novelty.
The manuscript topics fit enough to the journal scope.
A large number of previous researches by the authors and others have been discussed and those results have been compared to the results of the current research.
Sometimes the acronyms the first time that they are used in the document, the words are not written out with the short form placed in parentheses immediately after, some sentences are not clear due to grammatical errors, etc. At any rate, the topic is very important and it could have identifiable advance in knowledge.
The Conclusions are logical and based on the results of the research.
Author Response
Dear honorable reviewer,
Thank you for your constructive review and positive comments to our revised manuscript. Few sentences have been reformulated and several typos have been corrected now. A diff file capturing these changes is attached.
Your sincerely,
Dr. Octavian Andrei
On behalf of all co-authors

Reviewer 2 Report
The authors improve the text according to the reviewers remarks.
Author Response
Dear honorable reviewer,
Thank you once again for your constructive review and positive comments to our manuscript. Few sentences have been reformulated and several typos have been corrected now following the remarks from another reviewer. A diff file capturing these changes is attached fro your convenience.
Your sincerely,
Dr. Octavian Andrei
On behalf of all co-authors

This manuscript is a resubmission of an earlier submission. The following is a list of the peer review reports and author responses from that submission.
Round 1
Reviewer 1 Report
In the paper titled “Galileo L10 satellites: orbit, clock and signal-in-space performance analysis” the authors report on the performance of Galileo L10 satellites in terms of two orbital parameters, broadcast satellite clocks and signal in space (SiS) performance indicators based on all broadcast navigation data available.
The idea of the research is interesting and presents some novelty.
The manuscript topics fit enough to the journal scope.
A minimum number of previous researches by the authors and others have been discussed and those results have been compared to the results of the current research.
Nonetheless, in the paper, there are some aspects that need additional clarification, explanation, or revision before the paper would be ready for publication.
Reading the paper it is not clear the advantage in using the proposed approach compared to the existing approaches. The authors should highlight for the proposed approach the advantages and disadvantages respect the other approaches.
Regarding this phrase:
We fitted a standard linear trajectory model, i.e. linear trend plus periodical signals) to the data as depicted in Figure 7
The phrase is not understandable: there is a closing parenthesis but there is no open one. It is necessary to indicate what type of curve fitting (i.e. moving average, ...) was used and its characteristics (unweighted or weighted, ...).
What do the numbers on the abscissa scale of the graph in figure 7 represent? The label is missing.
The same happens in the graph in figure 8. In these graphs, the legend is also missing.
Regarding this phrase:
The analysis shows also peaks for 13.5 and 27 days. In addition, there are further daily and sub daily variations visible in Figure 6.
Insert the scale within days in the graph of figure 6
I do not agree with the approach that the authors have chosen for section 4. It would be better to have two separate sections: one for discussion and another, section 5 that should be added, for conclusions.
The conclusion explains what the current study found and should talk about what future studies should accomplish. Other main general findings are probably buried in the main section of the paper. I invite the authors to improve the paper conclusion in this sense.
Author Response
Dear honorable reviewers,
Thank you for your constructive review and valuable comments to our manuscript. We have revised the manuscript as advised, and we believe the result is an improved manuscript. You can find the revised manuscript in the attachment. Additionally, we provide answers to your specific points to
ensure that all your comments have been addressed and to provide justification where necessary.
Your sincerely,
Dr. Octavian Andrei
On behalf of all co-authors

Reviewer 2 Report
The article is clearly structured, to a large extent complete and very readable. The value of the article lies essentially in the independent validation of official data on the Galileo system’s performance. This provides a good justification for the publication oft the paper.
A few comments for improvement.
p 1.24: To facilitate identification, the satellites under consideration should be referred to as GSAT0219/E36, GSAT0220/E13, GSAT0221/E15, GSAT0222/E33 throughout the whole paper.
Figure 1: for completeness, the righthand scale (C – B –A) should be labelled 'orbital plane'.
Figure 3: the legend of the figure should include all used mathematical symbols (example: a : semi-major axis, e : eccentricity, etc.)
p9, 208: a reference to future work could be inserted at this point, referring to the presented work.
p9, 238: typo GSAT0129, should be GSAT0219.
Figure 10: even though all results are well within specification, there is a very pronounced difference between the SIS 95th percentile and SIS 99th percentile scales. The large difference seems to be due to a few, but large, outliers. This fact should not remain uncommented. If there is no plausible explanation at hand, that could also be stated.
Insert used unit in the caption (unit meter)
p 13, 298. The statement ‚Short-term correction variations are found to be within the one picosecond/second‘ is one of the central results of the study and is also mentioned in the abstract. However, I did not find any justification for this statement in the entire article. Figure 8 shows the general variation of the clock correction in unit ‚microsecond‘ and is therefore much too coarse for an assessment at the picosecond level. The given statement must be justified in one or another form, e.g. by including another graph or table in section 3.2.
Author Response

(The authors gave the same response as above.)

Reviewer 3 Report
Dear Authors, all comments are in the attached file

Author Response

(The authors gave the same response as above.)
